



**Junpeng Fu, Xingjie Chen \* and Zhaomin Lv** 

School of Urban Rail Transportation, Shanghai University of Engineering Science, Shanghai 201620, China
\* Correspondence: chenxingjie@sues.edu.cn

**Abstract:** As an important part of track inspection, the detection of rail fasteners is of great significance to improve the safety of train operation. Additionally, rail fastener detection belongs to small-target detection. The YOLOv4 algorithm is relatively fast in detection and has some advantages in small-target detection. Therefore, YOLOv4 is used for rail fastener status detection. However, YOLOv4 still suffers from the following two problems in rail fastener status detection. First, the features extracted by the original feature extraction network of YOLOv4 are relatively rough, which is not conducive to crack anomaly detection on rail fasteners. In addition, the traditional convolutional neural network has a larger number of parameters and calculations, which are difficult to run on the embedded system with low memory and processing power. To effectively solve those two problems, this paper proposes a rail fastener status detection algorithm based on MobileNet-YOLOv4 (M-YOLOv4). The edge features and texture features of rail fasteners are very important for rail fastener detection, and CSPDarknet53 cannot effectively extract the features of fasteners. The MobileNet is used to replace the CSPDarknet53 feature extraction network in the YOLOv4 algorithm, which can extract subtle features of rail fasteners and reduce the number of parameters and calculations of the algorithm. The experimental results show that the M-YOLOv4 algorithm has high detection accuracy and low resource consumption in rail fastener status detection. The false-alarm rate (FAR), missed-alarm rate (MAR), and error rate (ER) were 5.71%, 1.67%, and 4.24%, respectively, and the detection speed reached 59.8 fps. Compared with YOLOv4, the number of parameters and calculations were reduced by about 80.75% and 83.20%, respectively.

**Keywords:** rail fasteners; target detection; deep learning; YOLOv4; MobileNet

## 1. Introduction

In recent years, China's railway constructions have developed rapidly. The rapid growth of freight and passenger traffic has made the normal operation of railway facilities a top priority of railway maintenance work [1]. Rail fasteners, as an important part of the railway infrastructure [2], play a role in fixing rails and linking the rails to the rail sleepers. Their status can directly affect the safe operation of railways [3]. Common abnormalities in rail fasteners include fracture, displacement, dislodgement, etc. [4]. In the study of this paper, the rail fasteners with the above abnormal states are defined as abnormal rail fasteners, and the rail fasteners are only classified as a normal rail fastener and abnormal rail fastener. Damage to rail fasteners not only affects the comfort of passengers on the train but can also have a significant impact on the operation of the train in the event of serious damage [5].

Traditional rail fastener status detection is mainly carried out by hand; this method is not only inefficient, but also has huge security risks for inspectors [6]. Therefore, intelligent and automated methods of detection are required for the status detection of rail fasteners. In recent years, artificial intelligence has made major advances in the field of detection and control [7–10]. With the rapid development of machine learning and image-processing technology, its application in the field of rail fastener detection is becoming more and more widespread [11]. Traditional machine learning methods extract manually designed features and combine them with various classifiers to classify and detect targets. The

manually designed features can be divided into global features and local features. Although global features have good invariance and intuitive representation, they have a high feature dimension and are susceptible to interference such as lighting, rotation, noise, etc. Therefore, it is not conducive to the detection of rail fasteners. Local features are commonly used for rail fastener status detection. Fan et al. [12] proposed a linear local binary pattern coding method that considers the relationship between centroids and upper and lower neighborhoods, which is capable of efficiently representing the key components of rail fasteners. The experiments indicate that the method has good detection performance. However, local features suffer from the inability to extract features accurately for targets with smooth edges and sensitivity to directional information. Xu et al. [13] fused MB-LBP (multiblock local binary pattern) features and PHOG (pyramid histogram of oriented gradients) features to form a new image feature for training and introduced an Adaboost-SVM (support vector machine) classifier to classify the samples. The experimental results showed that this fused image feature method effectively improved the accuracy rate. When using methods that extract manually designed features and combine them with machine learning, the extracted features are not sufficient. There are still problems such as poor detection accuracy, weak robustness when processing images, and a high missed-alarm rate for defective-rail-fastener detection.

Deep learning algorithms have been developed in the field of feature extraction in target detection and are widely used for rail fastener status detection. These algorithms can be divided into two main categories; one is two-stage detection algorithms, which are based on candidate regions, such as a region-conventional neural network (R-CNN) [14], Fast R-CNN [15], Faster R-CNN [16], etc. Bai et al. [17] proposed a two-stage classification model based on the modified Faster R-CNN and the SVDD (support vector data description) algorithms. Through the verification, it was shown that the proposed method can improve the precision and accuracy of rail fastener detection. However, two-stage detection algorithms still have the problem of slow detection speed; the other is one-stage detection algorithms, which are end-to-end learning algorithms, such as single shot multibox detector (SSD) [18], you only look once (YOLO) [19], etc. Additionally, the YOLO algorithm is commonly used for rail fastener detection due to its fast detection speed. Lin et al. [20] proposed a rail fastener detection method based on the YOLOv3 algorithm, which enables the rapid detection of rail fastener defects. Lu et al. [21] proposed an improved YoloV3 algorithm to detect abnormalities of rail fasteners. The K-means++ algorithm was used to determine the anchor boxes. The network and the loss function were improved to increase detection efficiency. Qi et al. [22] proposed a new detection network architecture called MYOLOv3-Tiny. The depthwise and pointwise convolution were used and the backbone network was redesigned. The experiments showed the network achieved a higher detection precision and faster detection speed compared to state-of-the-art methods. Deep learning methods have the advantages of high learning ability, good network adaptability, and high robustness of the extracted features. Although the YOLO algorithm has a fast detection speed, it has poor detection results in small-target detection [23]. The YOLOv4 [24] algorithm uses the CSPDarknet53 [25] feature extraction network. Additionally, it also performs feature fusion and designs a multiscale prediction head, which effectively improves the detection effect of small-target detection. However, the YOLOv4 network still has the following problems. First, the rail fastener features extracted by the original feature extraction network are relatively rough, which is not conducive to rail fastener crack detection. In addition, the number of parameters and calculations of the algorithm is relatively high.

In order to further improve the accuracy of rail fastener status detection based on YOLOv4 and to reduce the number of parameters and calculations of YOLOv4, this paper proposes a rail fastener status detection algorithm based on MobileNet-YOLOv4 (M-YOLOv4). The algorithm uses a MobileNet [26] feature extraction network instead of a CSPDarknet53 network to extract rail fastener features, replacing the normal convolution operation in the feature extraction process with depth-separable convolutions. The MobileNet feature extraction network has more convolutional layers, so it can extract more

subtle features of rail fasteners. Through the rail fastener feature visualization experiment, it is proved that the features extracted by MobileNet are more detailed than those of CSPDarknet53, and the detection results of M-YOLOv4 are better than that of YOLOv4. In addition, the depthwise separable convolution can effectively reduce the number of parameters and calculations, and improve detection efficiency. Experiments show that the detection speed of the algorithm is greatly improved after using the depthwise separable convolution to replace the ordinary convolution. Additionally, the number of parameters and calculations is also greatly reduced.

This paper is organized as follows: Section 2 introduces the YOLOv4 algorithm, Section 3 introduces the MobileNet algorithm and the M-YOLOv4 algorithm proposed in this paper, Section 4 describes the experimental results, and it is concluded in Section 5.

## 2. YOLOv4 Algorithm [24]

The YOLO algorithm is an object recognition and localization algorithm. It is based on a deep neural network that unifies target-bounding box detection and category probability prediction as a regression problem. Additionally, it allows the neural network to predict the bounding box coordinates and the probability of belonging to a category directly from the original input image during prediction. It is an end-to-end target detection algorithm [19].

The YOLOv4 algorithm has been optimized in a number of ways compared to the previous generations of YOLO algorithms, effectively improving the detection of small objects [24].

The YOLOv4 algorithm improves the backbone feature extraction network. The CSPDarknet53 feature extraction network is introduced to replace the original Darknet-53 feature extraction network, based on the idea of the cross-stage partial network (CSPNet). The network solves the problem of gradient repetition in network optimization by dividing the feature mapping of the base layer into two parts and then merging them via a cross-stage hierarchy. The CSPDarknet53 network performs a total of five downsampling operations, of which the last three downsampling results are fed into the detection network as the output feature map. At the same time, the Mosaic data augmentation method was also proposed to augment the data. This method uses four images stitched together in a randomly scaled, cropped, and lined up manner, thus expanding the dataset, balancing the number of small and large targets, and allowing better robustness of the network [24].

Although the CSPDarknet53 feature extraction network in the YOLOv4 algorithm reduces the computational effort to some extent compared to the Darknet53 network, it has some impact on rail fastener feature extraction due to the fact that the network uses the normal convolution operation several times to extract features. The extraction of rail fastener features by the CSPDarknet53 is relatively rough, which is not conducive to the detection of rail fasteners. Additionally, a more ordinary convolution layer inevitably increases the number of calculations as the depth increases and generates a redundancy of parameters in rail fastener status detection. It also increases the complexity of the algorithm, lengthens the processing time of the algorithm, and fails to meet the requirements of real-time rail fastener detection.

## 3. Rail Fastener Status Detection Based on M-YOLOv4

### 3.1. MobileNet Network [26]

The MobileNet network is a lightweight deep neural network proposed by Google for embedded devices, whose core idea is depthwise separable convolution [26].

During the operation of ordinary convolution, only one feature is obtained from one convolution kernel. Additionally, to obtain more attributes, more filters are needed for stacking; the process is shown in Figure 1. Therefore, ordinary convolution will inevitably increase the number of parameters and calculations of the algorithm significantly.

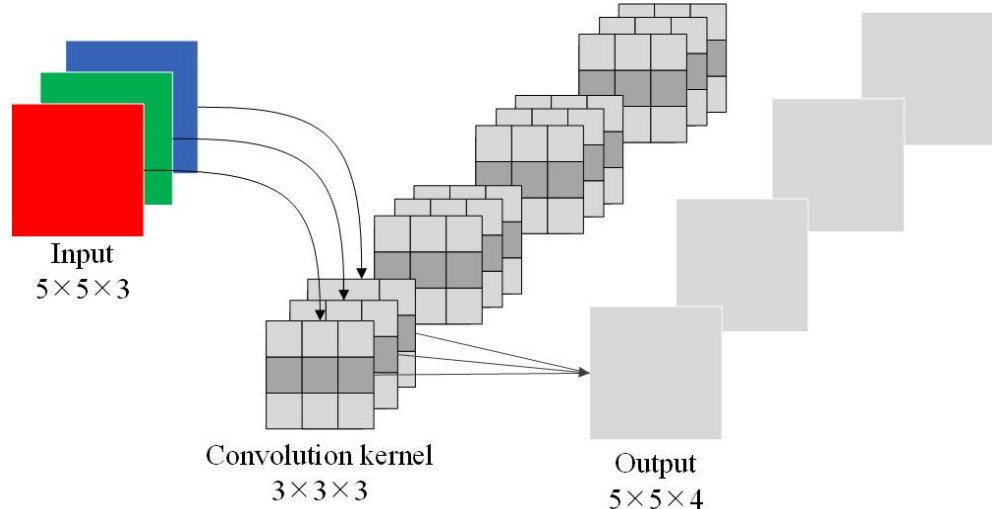

**Figure 1.** Schematic diagram of the ordinary convolution process.

The depthwise separable convolution decomposes the standard convolution into two parts: depthwise convolution and pointwise convolution. The depthwise convolution is responsible for filtering and acts on each channel of the input. The pointwise convolution is responsible for converting channels. The process of the depthwise separable convolution is illustrated in Figure 2. Since the convolution is split, MobileNet has more convolutional layers [26]. Therefore, more subtle features of rail fasteners can be extracted.

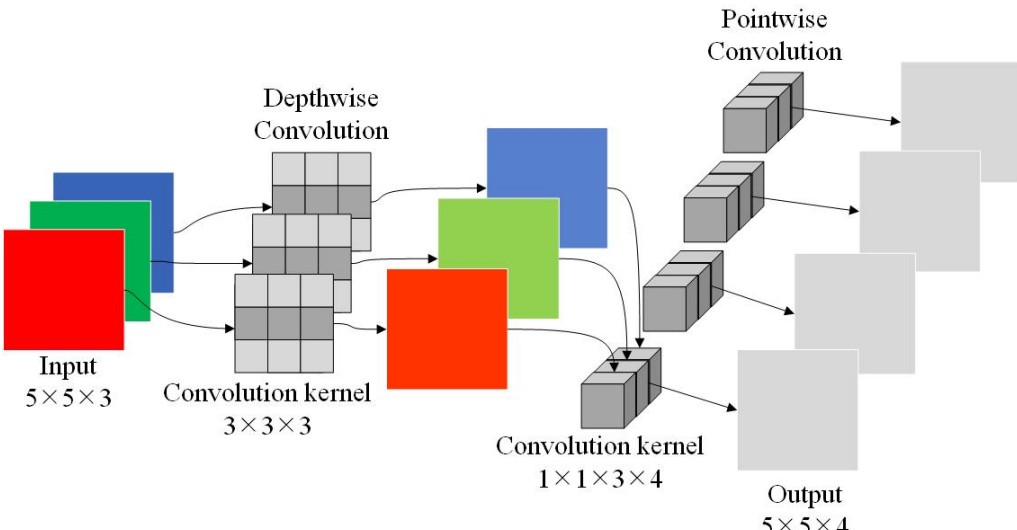

**Figure 2.** Schematic diagram of the depthwise separable convolution process.

It is assumed that the size of the input image is $W * H * C$, where $W$, $H$, and $C$ are the width, height, and number of channels of the image, respectively; the standard convolution kernel size is $K_W * K_H$. The output feature map size is $W_O * H_O * C_O$.

Thus, the number of parameters of the ordinary convolution operation is:

$$K_W * K_H * C * C_O \tag{1}$$

where $K_W$ and $K_H$ are the width and height of the convolution kernel, respectively.

The number of calculations is:

$$K_W * K_H * C * W_O * H_O * C_O \tag{2}$$

where $W_O$, $H_O$, and $C_O$ are the width, height, and number of channels of the output feature map, respectively.

Additionally, the number of parameters when using depthwise separable convolution is:

$$K_W * K_H * C + 1 * 1 * C * C_O \tag{3}$$

The number of calculations is:

$$K_W * K_H * C * W_O * H_O + 1 * 1 * C * W_O * H_O * C_O \tag{4}$$

Therefore, the ratio of the number of parameters is:

$$\frac{K_W * K_H * C + 1 * 1 * C * C_O}{K_W * K_H * C * C_O} = \frac{1}{C_O} + \frac{1}{K_W * K_H} \tag{5}$$

The ratio of the number of calculations is:

$$\frac{K_W * K_H * C * W_O * H_O + 1 * 1 * C * W_O * H_O * C_O}{K_W * K_H * C * W_O * H_O * C_O} = \frac{1}{C_O} + \frac{1}{K_W * K_H} \tag{6}$$

In order to facilitate the understanding and comparison, it is assumed that $C_O = C$ and considered that the commonly used convolution kernel size is 3×3. Thus, the amount of parameters and the amount of calculations of the depthwise separable convolution are reduced to nearly 1/9 of the conventional convolution [26].

*3.2. M-YOLOv4 Detection Algorithm*

For rail fasteners, in order to improve the accuracy of rail fastener status detection based on YOLOv4 and reduce the number of parameters and calculations of the algorithm, in this paper, we replace the backbone feature extraction network and propose the M-YOLOv4 algorithm.

The algorithm was divided into two parts: training and testing. In the training section, the labeled dataset was to be used to feed the network for training. The dataset contained different kinds of abnormal rail fastener images and normal rail fastener images for network training. First, the labeled rail fastener images were inputted; then, the images were resized to a fixed size by adding grayscale bars around the image. Then, the images were fed into the feature extraction network to extract features. In this paper, the MobileNet feature extraction network was used instead of the CSPDarknet53 network to extract the features of the input rail fastener images. Since the MobileNet uses more convolutional layers, the extracted features are more subtle. It can effectively help the extraction of rail fastener crack features.

After the images had undergone feature extraction and downsampling many times, the features of the last three layers of the network were fed as an output to the later operations. Since the abnormal state of rail fasteners includes fracture, displacement, and loss, different features need to be fused for better rail fastener status detection. In order to further extract fine-grained features, the deep feature map was upsampled and spliced with the feature map extracted from the upper layer, which is very helpful for the rail fastener status detection.

Finally, the detection result was output after passing through the Yolo detection layer. After the network had been trained several times, the network model with the best training results was saved and could be used for testing. Since the depthwise separable convolution was used instead of the ordinary convolution to perform the convolution operation on the image features, the M-YOLOv4 algorithm will have a greater reduction in the number of parameters and calculations compared to the YOLOv4 algorithm. In addition, since the feature extraction network uses more convolutional layers, it is beneficial for detecting cracks in rail fasteners. Additionally, because of the use of feature fusion, it is better for detecting different abnormal states of rail fasteners.

During the testing section, unlabeled rail fastener images were used to test. The rail fastener images were first resized to a fixed size, then fed into the trained network for detection. Finally, the localization and detection results of the rail fasteners were output in the image. The overall detection flow diagram of the algorithm in this paper is shown in Figure 3.

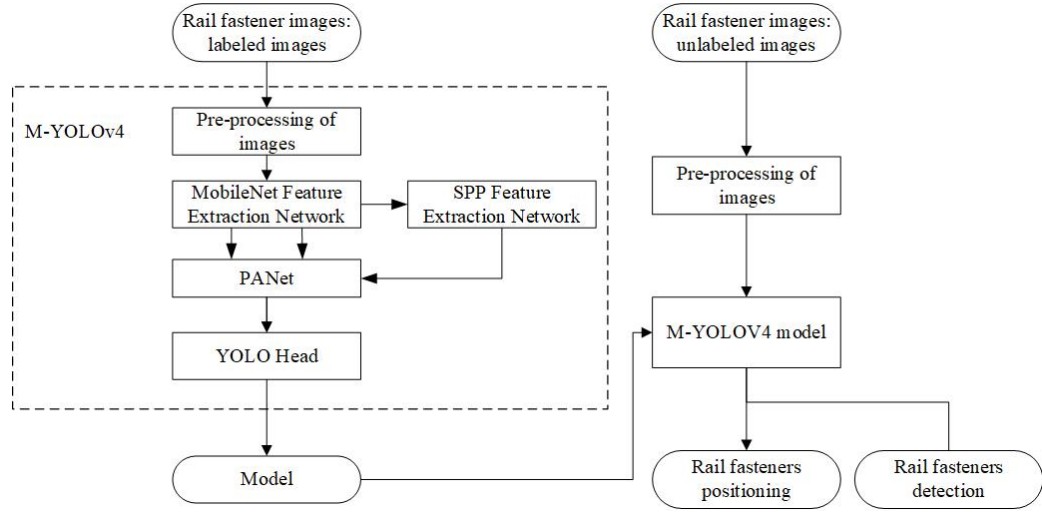

**Figure 3.** Flow diagram of M-YOLOv4.

The extracted features are more subtle due to replacing the CSPDarknet53 feature extraction network in the YOLOv4 algorithm with the MobileNet network. Additionally, the use of depthwise separable convolutions can greatly reduce the number of parameters and the number of calculations of the algorithm. Thus, the algorithm in this paper had better detection results of rail fastener status.

## 4. Experiments and Analysis

### 4.1. Experimental Setting

In the data preparation stage, the Labelimg was used to label the training images. Due to the special structure of rail fasteners, the bolts next to the rail fasteners will affect the detection results of the algorithm [6]. Therefore, when labelling the rail fastener pictures, the bolts were excluded in the images, and only the part of the rail fasteners in the images were labeled.

The experimental data were collected on a subway track. All images were acquired from the same angle with the same camera. A total of 500 original rail fastener images were collected. Due to the small number of original image samples, data augmentation operations such as rotation and mirroring were performed on the rail fastener images. After processing, there were a total of 1500 original images, of which 850 normal rail fastener images and 400 abnormal rail fastener images were selected as training data, 50 normal rail fastener images and 50 abnormal rail fastener images were used as the validation set, and 70 normal rail fastener images and 80 abnormal rail fastener images were used as test data.

The experimental platform was equipped with an Intel Core i7-10700 processor, an NVIDIA Quadro P2200 graphics card, and Micron 16 G/3200 MHz RAM, and the operation system was Windows 10. Python3.6, Pytorch1.2.0, Cuda10.0, and Cudnn7.4.1 were used to build the software environment for deep learning.

In order to obtain a better training effect, the algorithm performed Mosaic data enhancement in the image-preprocessing stage. The method read four pictures at a time, then flipped, zoomed, changed color gamut, etc., on the four pictures. Then, it arranged them in four directions, and finally combined the four pictures. After using this data augmentation method, the object detection background and small object datasets could be enriched.

In order to speed up the training, freeze backbone was adopted; that is, the pre-training weights already existing in the improved backbone network were first frozen, and more resources were applied to the training of the following parameters. Thus, the time and resource utilization could be greatly improved. The previously frozen parts were unfrozen after a while and trained together.

The batch size was set to 64 for the training phase and the initial learning rate was set to 0.001. To reduce the possibility of overfitting, a label smoothing method was used. A small penalty was applied to the classification accuracy so that the model could not predict too accurately to reduce the modelling of extreme cases around wrong answers to some extent. The label smoothing factor was set to 0.005. The image samples were trained for a total of 300 epochs. Figure 4 shows the comparison of loss curves, where the green curve represents the YOLOv4 algorithm loss curve and the red curve represents the M-YOLOv4 algorithm loss curve. The lower the loss value in training, the better the training result. It can be seen from the figure that the loss value of the M-YOLOv4 algorithm tended to be stable after 150 epochs and the loss value was slightly lower than that of the YOLOv4 algorithm.

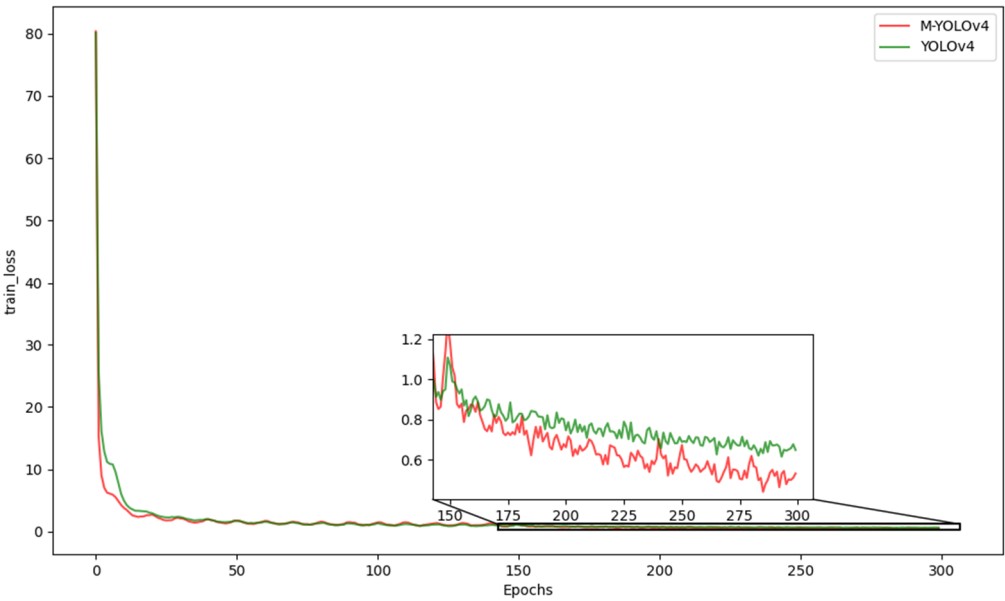

**Figure 4.** Comparison of loss curves.

*4.2. Feature Map Visualization*

In order to compare the features extracted by the CSPDarknet and the MobileNet, some feature maps of the two networks have been visualized separately. We randomly selected an image from the test set, then fed it into the YOLOv4 network and the M-YOLOv4 network for feature visualization, respectively, as shown in Figure 5. Figure 5a shows the original image of the rail fasteners, Figure 5b shows the feature visualization image extracted with the YOLOv4 network, and Figure 5c shows the feature visualization image extracted with the M-YOLOv4 network. As can be seen from Figure 5b,c, the M-YOLOv4 network is able to preserve the finer features of the image due to more convolutional layers. This has a beneficial effect on the crack detection of rail fasteners.

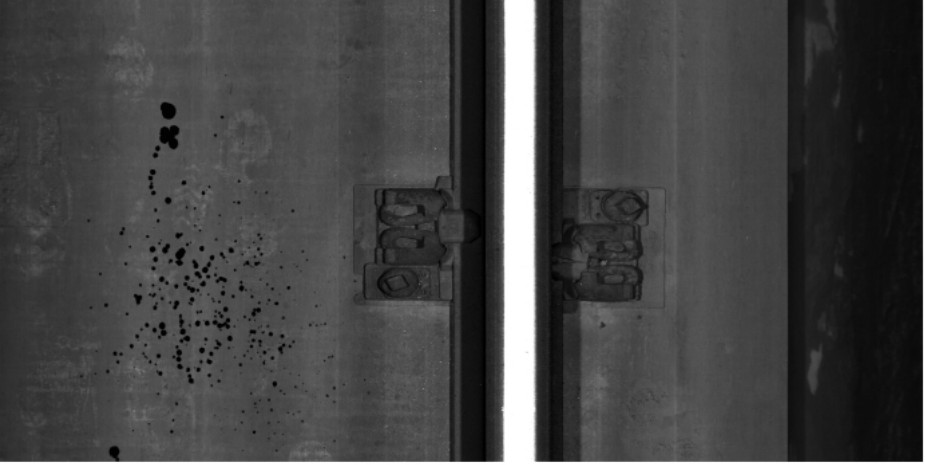

(a)Original image

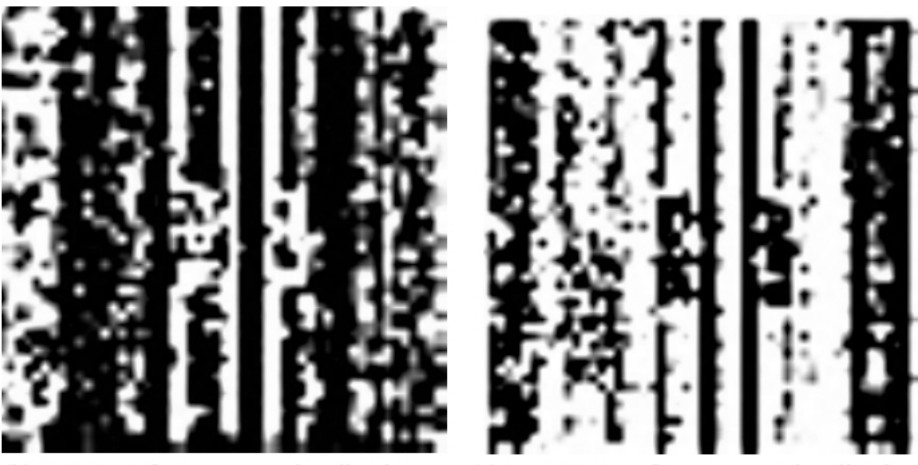

(b) YOLOv4 feature map visualization     (c) M-YOLOv4 feature map visualization

**Figure 5.** Feature map visualization.

### 4.3. Comparison and Results

4.3.1. Evaluation Indicators for the Testing of Rail Fasteners

The experiments compared the performance of the YOLOv4 algorithm with the M-YOLOv4 algorithm in terms of fastener detection indicators. This paper compared the false-alarm rate (FAR), the missed-alarm rate (MAR), and the error rate (ER) [6]. False alarm indicates that normal fasteners are detected as abnormal fasteners. Missed alarm indicates that abnormal fasteners are detected as normal fasteners. ER indicates the proportion of the sum of the number of normal rail fasteners detected as abnormal rail fasteners and the number of abnormal rail fasteners detected as normal rail fasteners in all detected rail fasteners. Additionally, the lower the false alarm rate, the missed alarm rate, and the Error rate are, the better the detection effect.

FPS (frames per second) refers to the number of pictures that the algorithm can detect per second, which is used to indicate the detection speed of the algorithm. The higher the value, the faster the algorithm detects. Since FPS can be affected by device configuration, this paper compares the results of both algorithms on the same device, thus reducing the impact of the device configuration on the detection speed.

Faster R-CNN, as one of the representatives of two-stage algorithms, had a large improvement in detection performance compared with R-CNN and Fast R-CNN. Therefore, the Faster R-CNN, YOLOv4, and M-YOLOv4 were compared through experiments. The comparison of detection results is shown in Table 1. It can be seen from Table 1 that the false alarm rate of the Faster R-CNN was 4.29%, the missed alarm rate was 1.67%, the error

rate was 3.33%, and the FPS was 4.28. Additionally, the false-alarm rate of the YOLOv4 algorithm was 9.62%, the missed-alarm rate was 3.33%, the error rate was 7.32%, and the FPS was 17.3. Additionally, the M-YOLOv4 algorithm was 5.71%, the missed-alarm rate was 1.67%, the error rate was 4.24%, and the FPS was 59.8. Although Faster R-CNN had slightly better detection results, its detection speed was much lower than YOLOv4 and M-YOLOv4, and could not meet the demand for real-time detection. Additionally, compared with YOLOv4, both the detection results and the detection speed of M-YOLOv4 were greatly improved. The features of rail fasteners extracted by the MobileNet network were more subtle, so the detection results of rail fasteners had a certain improvement.

**Table 1.** Comparison of detection results between YOLOv4 and M-YOLOv4.

|  | $D_N^N$ | $D_N^A$ | $D_A^A$ | $D_A^N$ | *FAR*/% | *MAR*/% | *ER*/% | **FPS** |
|---|---|---|---|---|---|---|---|---|
| Faster R-CNN | 201 | 9 | 118 | 2 | 4.29 | 1.67 | 3.33 | 4.28 |
| YOLOv4 | 188 | 20 | 116 | 4 | 9.62 | 3.33 | 7.32 | 17.3 |
| M-YOLOv4 | 198 | 12 | 118 | 2 | 5.71 | 1.67 | 4.24 | 59.8 |

Figure 6 shows the detection results of YOLOv4 and M-YOLOv4. There are some cracks in the rail fasteners in the image. The rail fastener on the left side of the rail in the image is a rail fastener with a crack, and the one on the right side of the rail is a normal rail fastener. The blue boxes in the image indicate that the algorithm considers the rail fastener to be a normal fastener and the red boxes indicate that the algorithm considers the rail fastener to be an abnormal fastener. Additionally, the number represents the probability that the algorithm considers the fastener to be a normal rail fastener or an abnormal rail fastener. Figure 6a shows the detection results of the YOLOv4 algorithm, and Figure 6b shows the detection results of the M-YOLOv4 algorithm. It can be seen from Figure 6 that YOLOv4 has the problem of a missing alarm for the cracked rail fastener, while M-YOLOv4 can reduce the missing alarm for cracked-fastener detection. Since the features extracted by MobileNet are more subtle, the M-YOLOv4 algorithm has a better detection effect on rail fasteners with cracks.

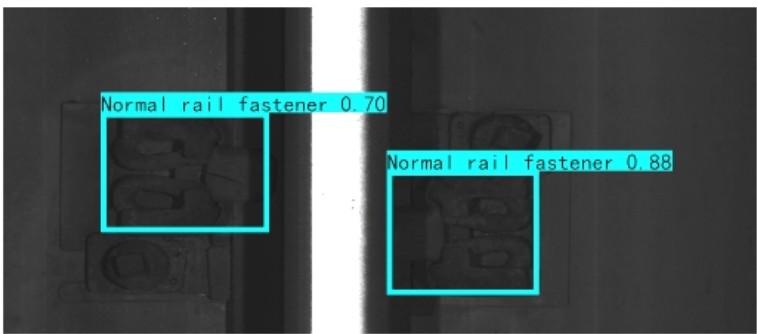

(a)YOLOv4 detection result

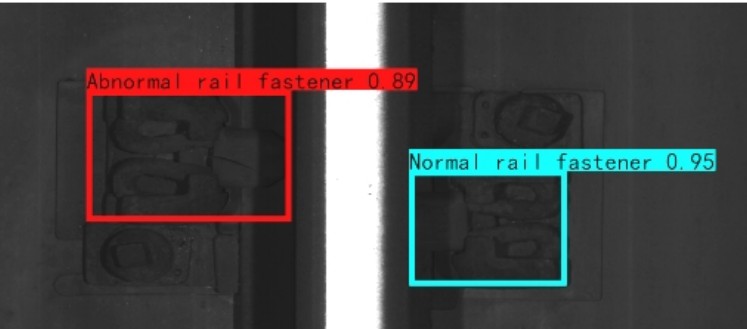

(b)M-YOLOv4 detection result

**Figure 6.** Detection results of rail fasteners with cracks.

To show the validity of the method, the labels of 10 normal rail fastener images and 10 abnormal rail fastener images were flipped and fed into training along with the training set. The rest of the settings were not changed in the training. The model was tested using the same test set. The comparison of results after adding noise are shown in Table 2. It can be seen from Table 2 that the false-alarm rate of Faster R-CNN increased to 5.24 and the error rate increased to 3.94 after adding noise, and the detection results became worse due to the noise. The addition of noise increased the M-YOLOv4 algorithm's false-alarm rate slightly to 6.22%, the missed-alarm rate was 0.83%, and the error rate was 3.31%. The overall performance of the algorithm remained basically unchanged, the slight fluctuations were caused by internal parameters, and the detection results were still good. Therefore, the algorithm proposed in this paper has certain anti-noise ability and good robustness.

**Table 2.** Comparison of results after adding noise.

|  | $D_N^N$ | $D_N^A$ | $D_A^A$ | $D_A^N$ | *FAR*/% | *MAR*/% | *ER*/% |
|---|---|---|---|---|---|---|---|
| Faster R-CNN | 201 | 9 | 118 | 2 | 4.29 | 1.67 | 3.33 |
| Faster R-CNN (label flipped) | 199 | 11 | 118 | 2 | 5.24 | 1.67 | 3.94 |
| M-YOLOv4 | 198 | 12 | 118 | 2 | 5.71 | 1.67 | 4.24 |
| M-YOLOv4 (label flipped) | 196 | 13 | 119 | 1 | 6.22 | 0.83 | 3.31 |

4.3.2. Performance Indicators

In order to intuitively show the optimization effect of the algorithm in this paper, the parameters (PRM) and floating point of operations (FLOPs) of the Faster R-CNN, YOLOv4, and the M-YOLOv4 were compared through experiments. The lower the value of the PRM and the FLOPs, the better. The comparison of PRM and FLOPs are shown in Table 3. It can be seen from Table 3 that the PRM of Faster R-CNN was about 0.0330 G, and the FLOPs was about 26.587 G. The PRM of the YOLOv4 was about 0.0639 G, and FLOPs was about 29.972 G, while the PRM of the M-YOLOv4 was about 0.0123 G, and the FLOPs was about 5.034 G. It can be seen that M-YOLOv4 had the least number of parameters and calculations. Additionally, the PRM and FLOPs of M-YOLOv4 were reduced by about 80.75% and 83.20%, respectively, compared with YOLOv4.

**Table 3.** Comparison of PRM and FLOPs.

|  | **PRM/G** | **FLOPs/G** |
|---|---|---|
| Faster R-CNN | 0.0415 | 121.670 |
| YOLOv4 | 0.0639 | 29.972 |
| M-YOLOv4 | 0.0123 | 5.034 |

It is easy to see that the M-YOLOv4 was much lower than the YOLOv4 in terms of parameters and calculations. This paper used depthwise separable convolution to replace the ordinary convolution in the feature extraction to reduce the complexity of the algorithm. Therefore, the detection speed was improved and the number of parameters and calculations of the algorithm were reduced.

**5. Conclusions**

This paper proposes a new rail fastener status detection algorithm named M-YOLOv4. In the feature extraction section, the MobileNet is used to replace the CSPDarknet53. Since the features of rail fasteners extracted by MobileNet are more detailed, the detection performance of rail fasteners can be improved to a certain extent. Moreover, the depthwise separable convolution is used instead of ordinary convolution, which reduces the number

of parameters and calculations of ordinary convolution, thereby improving the detection speed of the algorithm. The detection results of the YOLOv4 and the M-YOLOv4 are compared through experiments. The experimental results show that the M-YOLOv4 is better than the YOLOv4 in the performance of rail fastener status detection, and the detection speed is faster than the YOLOv4. However, as MobileNet has a more detailed feature extraction for rail fasteners, the algorithm will also detect fasteners with minor cracks as abnormal fasteners, thus potentially increasing the false-alarm rate. In addition, although M-YOLOv4 can identify abnormal rail fasteners, it is not yet able to make a detailed classification of different types of rail fastener abnormalities. More research will be conducted on the above two issues in the future.

**Author Contributions:** Conceptualization, J.F. and Z.L.; methodology, J.F.; software, J.F.; validation, J.F. and Z.L.; resources, Z.L. and X.C.; writing—original draft preparation, J.F.; writing—review and editing, Z.L. and X.C. All authors have read and agreed to the published version of the manuscript.

**Funding:** This research received no external funding.

**Data Availability Statement:** Not applicable.

**Conflicts of Interest:** The authors declare no conflict of interest.

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
