# Peer review of "Rail Fastener Status Detection Based on MobileNet-YOLOv4"

_electronics, doi:10.3390/electronics11223677_

Round 1

Reviewer 1 Report

line 195: reference missing

line207 spelling mistake 'status'

line 211: what data augmentation techniques were used?

line 260: spell check near error rate 

Author Response

  1. Comment: line195: reference missing.
  2. Reply: Thanks for your careful checks. We have corrected the mistake in the revised manuscript.

  1. Comment: line 207: spelling mistake ‘status’.
  2. Reply: We were really sorry for our careless mistakes. In our resubmitted manuscript, the typo is revised.

  1. Comment: line 211: what data augmentation techniques was used?
  2. Reply: Thank you for your comment. In the data augmentation section, we use rotation and mirroring to increase the number of images.

  1. Comment: line260: spell check near error rate.

     4.Reply: We feel sorry for our carelessness. In our resubmitted manuscript, the typo is revised.

Reviewer 2 Report

The paper presents a study of the task of position and quality detection of railway fasteners from their visible images. YOLO detector is used. It is modified by limiting the class of kernels to only separable ones, that reduces the number of trainable parameters. Numerical experiments are conducted with a training and testing set.

First, there is no original methods or their novel combinations presented in the paper. Both YOLO networks and reduction of parameters with separable kernels are well known. Their combination is not new either. The detailed description of this combination (which might reveal something new) is missing. There is no theoretical consideration of its effects, pros and cons, except a very straightforward estimation of the number of trainable parameters. This method is not compared against its rivals of other type. So, the paper has no theoretical meaning.

Second, looking from the practical side. The problem addressed in the paper can be generally described as the problem of quality control via the imagery. It is well-known classical problem of machine vision, in fact, one of the drivers of its very appearance. During the passed decades many works have set the standard of how to describe a numerical experiment to be feasible and conclusive. At first, the database should be collected so as to reflect the possible range of conditions (different illumination, different cameras, other varying factors, according to the problem). These factors should be enumerated and described also. Nothing of this kind here. Presumably, all images are taken from a single camera in a single pass, which makes all experimental conclusions completely unfounded. At second, 500 images for training set is a very small set for training CNN. It is small even for a simpler (in number of trainable parameters) Viola-Jones method. At third, the possible outcomes of the method are not enumerated and meaning of error numbers is not described. For example, what is FAR? False alarm that the fastener in image is cracked while it is normal? Or false detection of cracked fastener in the image with no fasteners at all? Or detection of cracker fastener in image with cracked fastener, but in wrong position?

Considering the above, the paper is inadequate for publication.

Author Response

  1. Comment: First, there is no original methods or their novel combinations presented in the paper. Both YOLO networks and reduction of parameters with separable kernels are well known. Their combination is not new either. The detailed description of this combination (which might reveal something new) is missing. There is no theoretical consideration of its effects, pros and cons, except a very straightforward estimation of the number of trainable parameters. This method is not compared against its rivals of other type. So, the paper has no theoretical meaning.
  2. Reply: We appreciate for your comment. Although the YOLO algorithm has a fast detection speed, it has poor detection results in small target detection. Compared with previous generations of YOLO, the YOLOv4 algorithm uses the CSPDarknet53 feature extraction network. And it also performs feature fusion and designs a multi-scale prediction head, which effectively improves the detection effect of small target detection. However, the YOLOv4 network still has the following problems. First, edge and texture features of rail fastener images are important for rail fastener detection, and the rail fastener features extracted by CSPDarknet53 cannot reflect the edge features of fasteners, which is not conducive to rail fastener crack detection. Therefore, MobileNet is used to replace the CSPDarknet53 and is more conducive to extracting the edge features of the rail fasteners. In addition, the amount of parameters and calculations of algorithm is relatively high. Therefore, depthwise separable convolution is used to replace the normal convolution, thus reducing the number of parameters and the calculations of the algorithm.

  1. Comment: At first, the database should be collected so as to reflect the possible range of conditions (different illumination, different cameras, other varying factors, according to the problem). These factors should be enumerated and described also. Nothing of this kind here. Presumably, all images are taken from a single camera in a single pass, which makes all experimental conclusions completely unfounded.
  2. Reply: Thank you for your comment. All images were acquired from the same angle with the same camera. It would be better to have more images of rail fasteners taken under different conditions. However, because of the limitation, the images of rail fasteners under different conditions could not be obtained. In the following works, rail fastener images under different conditions will be collected for experiments.

  1. Comment: At second, 500 images for training set is a very small set for training CNN. It is small even for a simpler (in number of trainable parameters) Viola-Jones method.
  2. Reply: Thanks for your comment. We expanded the number of images to 1500 for training, validation and testing of the algorithm by rotating and mirroring the 500 original images.

  1. Comment: At third, the possible outcomes of the method are not enumerated and meaning of error numbers is not described. For example, what is FAR? False alarm that the fastener in image is cracked while it is normal? Or false detection of cracked fastener in the image with no fasteners at all? Or detection of cracker fastener in image with cracked fastener, but in wrong position?

     4. Reply: Considering the Reviewer’s suggestion, we have added specific explanations of the evaluation metrics. Common abnormalities in rail fasteners include fracture, displacement, dislodgement, etc. In the study of this paper, the rail fasteners with the above abnormal states are defined as abnormal rail fasteners, and the rail fasteners are only classified as normal rail fastener and abnormal rail fastener. And False alarm indicates that normal fasteners are detected as abnormal fasteners. Missed alarm indicates that abnormal fasteners are detected as normal fasteners. ER indicates the proportion of the sum of the number of normal rail fasteners detected as abnormal rail fasteners and the number of abnormal rail fasteners detected as normal rail fasteners in all detected rail fasteners.

Reviewer 3 Report

The idea is interesting but the following issues should be considered during the preparation of the final version of the paper.

The title and abstract seem appropriate. The details about how the proposed mechanism works may be added to improve the attraction of the abstract.

The first part of the introduction has some problems with cited references. Please check the references and also check citations.

The main problem tackled in the paper has been elaborated well although the suggested method needs more elaboration by explaining more about details. 

With the existing material and also organization the novelty seems very narrow in comparison with YOLOv version. The modifications are explained well but as a novelty, we may expect to see more details.

I suggest adding more experiments. The existing results are required but not sufficient because they do not reflect all dimensions of the proposed method. 

Author Response

  1. Comment: The title and abstract seem appropriate. The details about how the proposed mechanism works may be added to improve the attraction of the abstract.
  2. Reply: We appreciate for your comment. The edge features and texture features of rail fasteners are very important for rail fastener detection, and CSPDarknet53 cannot effectively extract the features of fasteners. The MobileNet is used to replace the original CSPDarknet53 feature extraction network in YOLOv4 algorithm, which can extract subtle features of rail fasteners and reduce the number of parameters and calculations of the algorithm.

  1. Comment: The first part of the introduction has some problems with cited references. Please check the references and also check citations.
  2. Reply: Thanks for your careful checks. Based on your comment, we have checked the references and citations. The problems have been corrected in the revised manuscript.

  1. Comment: The main problem tackled in the paper has been elaborated well although the suggested method needs more elaboration by explaining more about details.
  2. Reply: Thank you for your comment. In rail fastener detection, the edge features and texture features of the image are important, and the CSPDarknet53 feature extraction network in YOLOv4 cannot extract sufficient edge features from rail fastener images, so the MobileNet is used to replace the CSPDarknet53 to extract features of rail fasteners, and experiments on feature visualization verified that the rail fastener features extracted by MobileNet are finer.

  1. Comment: With the existing material and also organization the novelty seems very narrow in comparison with YOLOv version. The modifications are explained well but as a novelty, we may expect to see more details.

     4. Reply: Thanks for your comment. Although YOLOv4 shows a considerable improvement in the detection of small targets, in rail fastener detection, the features of rail fasteners extracted by the CSPDarknet53 feature extraction network are not sufficient, which leads to its poor detection results. Therefore, MobileNet is used to replace CSPDarknet53, and the feature visualization experiment proves that the MobileNet extracts finer features, which is beneficial to rail fastener detection.

Reviewer 4 Report

Dear Authors,

Please find the attached file for your reference. Please update the paper based on the comments and resubmit it. 

Regards 

Author Response

  1. Comment: Please add experiment results in the abstract。

1.Reply: Thank you for your suggestion. We have added the experiment results in the abstract in the revised manuscript.

  1. Comment: In the introduction, the authors mentioned ‘our country. However, the country name is not mentioned. Please update these details.
  2. Reply: We appreciate for your suggestion. The country referred to in the paper is China and we have updated it in the revised manuscript.

  1. Comment: Something wrong with the citation. Please check it. Lines 24 to 53.
  2. Reply: We feel sorry for our carelessness. Based on your comment, we have checked the citations and corrected the mistakes.

  1. Comment: Please write the full form of MB-LBP, PHOG, SVM, and SVDD.
  2. Reply: Thanks for your suggestion. The full form of MB-LBP (Multi-block Local Binary Pattern), PHOG (Pyramid Histogram of Oriented Gradients), SVM (Support Vector Machine) and SVDD (Support vector data description) have been added in the revised manuscript.

  1. Comment: Please add paper contributions.
  2. Reply: Thanks for your comment. We have added author contributions at the end of the paper.

  1. Comment: Please add paper organization.
  2. Reply: We appreciate for your suggestion. This paper is organized as follows: Section 2 introduces the YOLOv4 algorithm, Section 3 introduces the MobileNet algorithm and the M-YOLOv4 algorithm proposed in this paper, Section 4 describes the experimental results, and it is concluded in Section 5.

  1. Comment: Please check the Section 2 heading alignment. Please move to the next page for better readability.
  2. Reply: Thanks for your suggestion. Based on your comment, we have made adjustments to the Section 2 heading.

  1. Comment: Please add a full stop at the end of the figure captions.

8.Reply: We sincerely appreciate the valuable comments. We have added the full stop at the end of the figure captions and the table captions.

  1. Comment: Please check line 195.
  2. Reply: Thanks for your careful checks. We have corrected the mistake in the revised manuscript.

  1. Comment: Please check the typos and writing style.
  2. Reply: We feel sorry for our carelessness. In our revised manuscript, the typo is corrected. Thanks for your correction.

  1. Comment: Please check the Table 1 alignment.
  2. Reply: Thanks for your checks. We have corrected the mistake in the revised manuscript.

  1. Comment: Please use the word Table instead of table.
  2. Reply: Thanks for your careful checks. We have replaced ‘table’ with ‘Table’.

  1. Comment: Please use the same format for MobileNet (Please avoid the usage of mobilenet, line 283).
  2. Reply: We sincerely thank the reviewer for careful reading. As suggested by the reviewer, we have corrected the ’mobilenet’ into ‘MobileNet’.

  1. Comment: Please define FPS in line 299.
  2. Reply: Thanks for your suggestion. We have added the definition of FPS in the revised manuscript, and FPS (Frames Per Second) refers to the number of pictures that the algorithm can detect per second, which is used to indicate the detection speed of the algorithm.

  1. Comment: If possible, please add existing object detection algorithms for comparison. The authors used only one object detection algorithm in the current form, making it difficult to claim their approach performance better. (Example: R-CNN [14], Fast R-CNN [16], Faster- RCNN [16] etc,.).
  2. Reply: We appreciate for your valuable suggestion. Considering that Faster R-CNN is one of the representatives of one stage algorithm, the detection results of the Faster R-CNN, YOLOv4 and M-YOLOv4 are compared through experiment. The results show that the False alarm rate of the Faster R-CNN is 4.29%, the Missed alarm rate is 1.67%, and the Error rate is 3.33%, which is slightly better than M-YOLOv4, but its FPS is only 4.28, which is much lower than M-YOLOv4 and cannot meet the demand of real-time detection.

  1. Comment: Please add a citation for the CSPDarknet53 network.

   16. Reply: Thank you for your suggestion. We have cited related literature about CSPDarknet53 network in the place of the revised manuscript.

Round 2

Reviewer 2 Report

The authors tried to correct the paper according to the comments. However the paper major drawbacks are still in place, they cannot be removed by small cosmetic additions.

As I said in the previous review, there can be two ways to produce a quality work on the topic, which may deserve publishing in high-ranked journal. First way is the theoretical advance. The authors claim such an advance is using alternative feature extraction mechanism for YOLO architecture. Well, this can be a theoretical advance. But it should be proven, that this is a real advance. What are its strengths? Better convergence and less calculation, as authors say - let it be. What are its weaknesses? No weaknesses are pointed out, but I can hardly believe that there are none at all. What are its limitations? The strengths of better convergence and less calculation are demonstrated for single small database. As I said before, in such case the conclusions are just unfounded. As a result, the proposal to replace feature extractor in YOLO has no general meaning, there is no evidence, that it works wherever outside of the database of 500 rail images.

Second way to make a useful work is to develop an application. There is a definite application field: detecting abnormal rail fasteners. How can I be sure that the authors succeeded in developing a practically useful method of abnormality detection? All images were taken in a single experiment with single camera. One takes another camera and everything will fall down. One uses another illumination (daylight, or night, or bystander illumination source) and everything will shut down. One move camera with different speed and everything will crack down. The system developed for such narrow conditions has no practical sense. The authors say "In the following works, rail fastener images under different conditions will be collected for experiments." Great. Collect data in various conditions, as it is always in practice, and I assure, there will be many new real problems. Solve them, describe how, and this will be really interesting work. Now it is not.

Considering the above, the paper is still not suitable for publication.

Author Response

  1. Comment: The authors claim such an advance is using alternative feature extraction mechanism for YOLO architecture. Well, this can be a theoretical advance. But it should be proven, that this is a real advance. What are its strengths? Better convergence and less calculation, as authors say - let it be. What are its weaknesses? No weaknesses are pointed out, but I can hardly believe that there are none at all. What are its limitations? The strengths of better convergence and less calculation are demonstrated for single small database. As I said before, in such case the conclusions are just unfounded. As a result, the proposal to replace feature extractor in YOLO has no general meaning, there is no evidence, that it works wherever outside of the database of 500 rail images.
  2. Reply: Thanks for your comment. Edge and texture features of rail fastener images are important for rail fastener detection. And the strengths of M-YOLOv4 are as follows: Firstly, compared to CSPDarknet53, MobileNet has a more detailed extraction of edge features for track fasteners, which is more beneficial for rail fastener detection; Secondly, compared to YOLOv4, M-YOLOv4 has a lower number of parameters and calculations, and an increased detection speed. The weaknesses of M-YOLOv4 are as follows: Firstly, as MobileNet has a more detailed feature extraction for rail fasteners, the algorithm will also detect fasteners with minor cracks as abnormal fasteners, thus increasing the false alarm rate; Secondly, although M-YOLOv4 can identify abnormal rail fasteners, it is not yet able to make a detailed classification of different types of rail fastener abnormalities. It would be better to have more images of rail fasteners taken under different conditions. However, due to factors such as experimental funding, we are currently unable to capture images of rail fasteners under various conditions and can only use existing images for our experiments.

  1. Comment: How can I be sure that the authors succeeded in developing a practically useful method of abnormality detection? All images were taken in a single experiment with single camera. One takes another camera and everything will fall down. One uses another illumination (daylight, or night, or bystander illumination source) and everything will shut down. One move camera with different speed and everything will crack down. The system developed for such narrow conditions has no practical sense. The authors say "In the following works, rail fastener images under different conditions will be collected for experiments.” Great. Collect data in various conditions, as it is always in practice, and I assure, there will be many new real problems. Solve them, describe how, and this will be really interesting work. Now it is not.

2. Reply: Thank you for your comment. Different cameras capture different picture pixels. The camera we use captures pictures with 4096×2048 pixels, and M-YOLO can detect all the pictures of rail fasteners with camera pixels higher than this pixel, the retained features are more complete, and M-YOLOv4 can detect these rail fastener images. Moreover, the rail fastener images taken under conditions of sufficient light source are more favorable for detection, while detection becomes difficult under conditions of insufficient light source, such as at night. We have used the camera with light source to capture the images of rail fasteners, so the detection results are better. For low-speed urban rail transit, M-YOLOv4 is more suitable for detection, while for high-speed trains, the collected images cannot be transmitted in real time due to the limitation of equipment communication, and therefore cannot be detected in real time. And due to the limitation of experimental funds and other reasons, we cannot collect the images of rail fasteners under various conditions now, and we can only study and experiment with the existing images.

Reviewer 3 Report

The authors have replied to all of my comments. 

Author Response

Thank you for your help in improving the quality of the article.

Reviewer 4 Report

Dear Authors,

Thank you for addressing all my comments, and I don't have any further concerns about your paper. 

Regards 

Author Response

(The authors gave the same response as above.)
